# Chronic stimulation of group II metabotropic glutamate receptors in the medulla oblongata attenuates hypertension development in spontaneously hypertensive rats

**Julia Chu-Ning Hsu, Shin-ichi Sekizawa\*, Ryota Tochinai, Masayoshi Kuwahara\***

Department of Veterinary Pathophysiology and Animal Health, Graduate School of Agricultural and Sciences, The University of Tokyo, Tokyo, Japan

\* akuwam@mail.ecc.u-tokyo.ac.jp (MK); a-ssekiz@mail.ecc.u-tokyo.ac.jp (SS)

**Data Availability Statement:** All relevant data are within the manuscript and its Supporting Information files.

## Abstract

Baroreflex dysfunction is partly implicated in hypertension and one responsible region is the dorsal medulla oblongata including the nucleus tractus solitarius (NTS). NTS neurons receive and project glutamatergic inputs to subsequently regulate blood pressure, while G-protein-coupled metabotropic glutamate receptors (mGluRs) play a modulatory role for glutamatergic transmission in baroreflex pathways. Stimulating group II mGluR subtype 2 and 3 (mGluR2/3) in the brainstem can decrease blood pressure and sympathetic nervous activity. Here, we hypothesized that the chronic stimulation of mGluR2/3 in the dorsal medulla oblongata can alleviate hypertensive development via the modulation of autonomic nervous activity in young, spontaneously hypertensive rats (SHRs). Compared with that in the sham control group, chronic LY379268 application (mGluR2/3 agonist; 0.40 μg/day) to the dorsal medulla oblongata for 6 weeks reduced the progression of hypertension in 6-week-old SHRs as indicated by the 40 mmHg reduction in systolic blood pressure and promoted their parasympathetic nervous activity as evidenced by the heart rate variability. No differences in blood catecholamine levels or any echocardiographic indices were found between the two groups. The improvement of reflex bradycardia, a baroreflex function, appeared after chronic LY379268 application. The mRNA expression level of mGluR2, but not mGluR3, in the dorsal medulla oblongata was substantially reduced in SHRs compared to that of the control strain. In conclusion, mGluR2/3 signaling might be responsible for hypertension development in SHRs, and modulating mGluR2/3 expression/stimulation in the dorsal brainstem could be a novel therapeutic strategy for hypertension via increasing the parasympathetic activity.

## Introduction

Hypertension is a leading risk factor for cardiovascular diseases worldwide [1], and blood pressure (BP) control is important in reducing the risk of these illnesses [2]. Hypertensive patients

**Funding:** The authors received no specific funding for this work.

**Competing interests:** The authors have declared that no competing interests exist.

are consistently cautioned to exercise regularly, manage their diet, and take antihypertensive medication [1]. Although hypertension treatment involves various antihypertensive drugs including alpha- and beta-adrenergic blockers, diuretics, angiotensin converting enzyme inhibitors, angiotensin receptor blockers, vasodilators, and/or calcium-channel blockers [3], these remain unsatisfactory.

Baroreflex function properly maintains heart rate (HR) and BP within a physiological range to support the whole-body system for active daily life [4], and its pathways are greatly involved in medulla oblongata areas. Among which, the nucleus tractus solitarius (NTS) is the first synaptic site to receive afferent information from glutamatergic peripheral baroreceptors and airway receptors [5, 6]. NTS neurons send glutamatergic projections to neurons located in the ipsilateral caudal ventrolateral medullary depressor area (CVLM), which then send inhibitory gamma-aminobutyric acid (GABA)ergic projections to neurons located in the ipsilateral ventrolateral medullary pressor area (RVLM) [7]. Activation of second- and higher-order baroreceptor NTS neurons excites the CVLM neurons, which in turn inhibit the sympathoexcitatory RVLM neurons. These actions finally elicit depressor and bradycardic responses. G-protein-coupled metabotropic glutamate receptors (mGluRs), which provide acute and long-term modulation of glutamatergic transmission in various neural networks [8–10], are also present in the medulla oblongata, particularly in the NTS [11, 12]. Microinjection of mGluR modulators into the dorsal area of medulla oblongata has been previously attempted with Sprague-Dawley or Wistar rats for short-term administration. Of these modulators, agonists for group II mGluR subtypes 2 and 3 (mGluR2/3) could acutely change HR, BP, and sympathetic nervous activity [7, 13–16], but these results show discrepancies.

Spontaneously hypertensive rats (SHRs) have been widely used as a model animal of essential hypertension for studies on the mechanism of this hypertension [17]. The HR and BP of SHR and Wistar Kyoto rats (WKYs) were nearly the same until the age of 2 weeks [18]. Thereafter, however, the HR of SHRs becomes faster from the age of 3 weeks and the BP of SHRs becomes slightly higher at the age of 4 weeks compared to WKYs [19]. Meanwhile, the firing rates of the extracellular units of RVLM neurons are higher in adult SHRs than WKYs *in vivo* [20]. The electrophysiological properties of RVLM neurons and their responses to angiotensin II differ between WKYs and SHRs, even in the neonatal stage [21]. mGluRs, especially postsynaptic ones, do not exhibit any function at younger (juvenile) stages in some cases [22–24] but regulate various neuronal functions at different developmental stages [25–34]. Furthermore, mGluR2/3 agonists hyperpolarize neurons in normotensive rats [35], while the agonists effects seemed to be attenuated in adult SHRs (authors' observation). In this work, we hypothesized that the chronic stimulation of mGluR2/3 in the dorsal medulla oblongata, especially NTS, of juvenile SHRs might alleviate the development of hypertension by modulating the autonomic nervous activities. For hypothesis testing, a selective mGluR2/3 agonist, LY379268, was added to the extracellular cerebrospinal fluid of the brainstem with a mini-osmotic pump for the chronic agonist treatment. HR and BP were measured by the tail-cuff method throughout the developmental stages of hypertension, and autonomic nervous activity was evaluated by the power spectral analysis of heart rate variability (HRV) using a radio-telemetry system.

## Materials and methods

### Animals

All experimental protocols were approved by The Animal Care and Use Committee of the University of Tokyo (No. P17-033). All animals were managed according to the Guidelines for the Care and Use of Laboratory Animals established by the Graduate School of Agriculture and Life Sciences at the University of Tokyo.

Four-week-old male SHRs and WKYs were purchased from Charles River Laboratories Japan, Inc. (Yokohama, Japan) and housed in a temperature-controlled room (24 ± 1˚C) under an automatic controlled lighting (light on: 8:00–20:00) with access to food and water *ad libitum*.

## Dorsal hindbrain mGluR2/3 treatment

Isoflurane was used as anesthetic (Pfizer Japan Inc., Tokyo, Japan), and the following surgical procedures were performed under a specific level of anesthesia, i.e., the absence of withdrawal reflexes. Animals aged 6 weeks were implanted with a mini-osmotic pump (ALZET Model 2006, DURECT Corporation, Cupertino, CA, United States) that can continuously deliver solutions for 6 weeks of treatment, and the pump was removed at the end of the treatment. For the implantation, the lateral cervical space was opened, and a catheter (external diameter 0.61 mm, internal diameter 0.28 mm) was inserted in the cranial cavity through the foramen magnum. Its tip was located near the caudal end of the medulla oblongata, and its other end was connected to the mini-osmotic pump in the subcutaneous area of the back, which was filled with LY379268 ((1R,4R,5S,6R)-4-Amino-2-oxabicyclo[3.1.0]hexane-4,6-dicarboxylic acid) (0.40–40.0 μg/day) at least 60 hours prior to implantation. Sham surgery without mini-osmotic pump was performed on control rats.

## Measurement of HR and BP

The HR and BP of conscious animals were measured with the tail-cuff apparatus (BP-98AL, Softron Co., Ltd., Tokyo, Japan) at 3 days before surgery and every week in 7- to 18-week-old rats. Details of the operating procedures can be found in previous paper [36].

## Echocardiographic and renal ultrasonographic measurements

Echocardiography and renal ultrasonography tests were performed in 18-week-old SHRs under 2% isoflurane anesthesia in air with flow rate of 1 l/min by using a preclinical imaging system (Vevo 3100, FujiFilm VisualSonics, Toronto, ON, Canada) and a linear array transducer (MS-550S, FujiFilm VisualSonics, Toronto, ON, Canada). Echocardiography was recorded using the preclinical imaging system [37]. Image analysis was performed for left-ventricular short-axis, left-ventricular inflow waveform, and mitral valve septal tissue waveform. Cardiac function parameters such as HR, ejection fraction (EF), and cardiac output (CO) were calculated using an analysis software (Vevo LAB, FujiFilm VisualSonics, Toronto, ON, Canada). The parameters of renal arteries on both sides, especially resistive index (RI) and pulsatility index (PI) that indicate the severity of renal diseases, were also examined and calculated. A RI higher than 0.75 or a PI higher than 1.55 indicates chronic renal failure [38].

## Implantation of telemetry device for electrocardiography recording

In addition to the mini-osmotic pump, an ECG telemetry device (ATE-01S, Softron Co., Ltd., Tokyo, Japan) was implanted in the backs of 6-week-old SHRs under isoflurane anesthesia. Paired wire electrodes of the transmitter were subcutaneously placed on the dorsal and ventral thorax to record the apex-base lead ECG. Recordings were started a week after surgery with a signal receiving board (ATR-1001, Softron Co., Ltd., Tokyo, Japan) placed underneath each cage in the chamber (MIR-554, Panasonic, Japan) with standardized temperature (24˚C) and lighting (8:00–20:00). An ECG processor system (Softron Co., Ltd., Tokyo, Japan) was used to continuously record ECG signals.

## HRV analysis

Power spectral analysis of HRV was performed as previously described [39, 40]. Time- and frequency-domain methods were used to assess autonomic nervous activity. The time-domain analysis was based on R-R intervals for calculating standard deviation (SD) and coefficient of variation (CV), which are regarded as the indices of parasympathetic activity [40]. In the frequency-domain method, power spectral components were primarily classified into low (LF; 0.04 to 1.0 Hz) and high (HF; 1.0 to 3.0 Hz) frequency ranges as different elements of autonomic nervous activities [39]. The normalized power spectral components of low frequency and high frequency (LFnu and HFnu, respectively) were also calculated to diminish the influence of very low frequency and highlight the interaction between sympathetic and parasympathetic nerves [41, 42]. LF is affected by sympathetic and parasympathetic nervous activities, HF is as an index of the parasympathetic nervous activity, and the ratio of LF to HF (LF/HF) is as an index of the balance of autonomic nervous system [39].

## Measurement of catecholamine concentration

In brief, 12-week-old SHRs were decapitated under isoflurane anesthesia, and blood samples were collected from the left superior vena cava. The catecholamine concentration in blood serum was assessed with an ELISA kit (Cat Combi ELISA RUO EIA-4309R, DRG Instruments GmbH, Marburg, Germany) in accordance with the manufacturer's instructions.

## Assessment of baroreflex sensitivity

After dorsal hindbrain mGluR2/3 treatment was completed at 12 weeks of age, implants were removed, and SHRs were subjected to invasive catheterization. They were anesthetized with an intraperitoneal injection of urethane (1.5 g/kg) dissolved in distilled water. The animals were placed in the supine position, and then the following surgical procedures were performed after confirming the abolition of pain reflexes induced by pinching their paws. A polyethylene catheter was inserted into the left femoral vein for intravenous administration of pharmacological agents. The left femoral artery was also cannulated to measure arterial BP.

Arterial BP was tracked via the catheterized femoral artery with a catheter-transducer system (Nihon Kohden, Tokyo, Japan) connected to a computer acquisition system (Softron Co., Ltd., Tokyo, Japan) during the entire measurement. The mean arterial pressure (MAP) was calculated from measured systolic and diastolic BP. Limb lead II electrocardiogram was recorded with needle electrodes. After 5 min of basal control recording, intravascular injections of phenylephrine (PE; 21 µg/kg) or sodium nitroprusside (SNP; 50 µg/kg) were performed through the catheter of the femoral vein [43, 44]. Cardiovascular waveforms were then recorded for 15 min. After eliminating interference, HR and MAP were measured with averaged of each recording minute.

Baroreflex sensitivity was evaluated with changes in MAP (ΔMAP) and corresponding HR reflex (ΔHR) at the peak responses to PE and SNP application [43, 45]. The baseline data for comparing peak responses were calculated as the average values over the 5-min period of basal value recording. PE and SNP increase or decrease MAP accordingly, so HR responses are defined as bradycardia and tachycardia reflex, respectively [43, 46].

## RNA isolation and quantitative real-time Polymerase Chain Reaction (PCR)

Sham control and LY379268 treated groups of SHRs and WKYs were decapitated at 21 weeks of age under deep isoflurane anesthesia. Sham control groups of SHRs had developed sufficient

hypertension at that age. Whole medulla oblongata specimens were dissected, and total RNA was isolated using TRIzol reagent (Gibco-BRL, Grand Island, NY, United States). First-strand cDNA was synthesized using SuperScript IV VILO Master Mix with ezDNase Enzyme (Invitrogen, Carlsbad, CA, United States). With cDNA as a template, real-time PCR was performed with THUNDERBIRD SYBR qPCR Mix (Toyobo, Osaka, Japan) and LightCycler (Roche, Mannheim, Germany). The following primers for real-time PCR were designed based on published sequences: rat glyceraldehyde 3-phosphate dehydrogenase (GAPDH; an internal control) forward primer, 5′-TCA CCA TGG AGA AGG -3′; reverse primer, 5′-GCT AAG CAG TTG GTG CA-3′; rat mGluR2 forward primer, 5′-CGT GAG TTC TGG GAG AG-3′; reverse primer, 5′- GCG GAC CTC ATC GTC AGT AT-3′; rat mGluR3 forward primer, 5′-GTG GTC TTG GGC TGT TTG TT-3′; reverse primer, 5′-GCA TGT GAG CAC TTT GT-3′.

## Statistical analysis

All data are expressed as the mean ± SEM. HRV data were averaged from 24-hour recordings, or 12-hour recordings of either light or dark photoperiod. Prism 8 software (Graphpad Software Inc., San Diego, CA, United States) was used for all statistical analysis. Dose-dependent hypertensive effect was evaluated by one-way ANOVA, followed by Tukey's test. The Kaplan-Meier curve and the log-rank test were used to examine SHR mortality rate difference among different doses of LY379268 treatment. The dose and number of deaths after LY379268 treatment were used to calculate the 50% lethal dose ($LD_{50}$) (Quest Graph™ $LD_{50}$ Calculator, AAT Bioquest, Inc., Sunnyvale, CA, United States). HR and BP between WKYs and SHRs during LY379268 treatment were evaluated through two-way ANOVA, followed by Tukey's test for multiple comparisons. Statistical analyses were performed using unpaired t-test to examine other types of data. Significant differences were considered at $p < 0.05$.

## Results

Five groups of 6-week-old SHRs received five different doses of LY379268. Within a week after surgical operation, systolic BP (SBP) seemed to be reduced for the 0.40 μg/day dose, but had the tendency of getting close to control values with higher doses. However, no significant difference was found between any two doses ($P = 0.30$) (Fig 1A). Kaplan-Meier survival with a follow-up period revealed that the SHRs receiving 0.40 μg/day LY379268 showed the best survival rate without death as confirmed by the log-rank test (Fig 1B, $P < 0.05$), although LY379268 exhibited systemic toxicity (e.g., losing weight dramatically and less capability of action to meet endpoints for euthanasia) ($LD_{50} = 4.63$ μg/day) when administered to the dorsal medulla oblongata of SHRs (Fig 1C). Therefore, 0.40 μg/day was selected for following studies to achieve the same level of effect on BP.

## Effects of LY379268 (0.40 μg/day) on BP and HR in hypertensive development

Fig 2 shows time course changes of SBP, diastolic BP (DBP), and heart rate (HR) in 6- to 18-week-old SHRs. SBP and DBP increased with age in SHRs but not in WKYs regardless of treatment (Fig 2A and 2C). After LY379268 (mGluR2/3 agonist) treatment ended at 12 weeks of age, the agonist effect on SBP was evident in SHRs (sham control vs. agonist, 203.9 ± 2.1 vs. 160.4 ± 14.7 mmHg; $P < 0.001$) but not in WKYs (sham control vs. agonist, 123.5 ± 3.1 vs. 111.4 ± 5.3 mmHg; $P = 0.62$) (Fig 2B). This effect persisted at 18 weeks of age, 6 weeks after the absence of treatment (sham control vs. agonist, 202.9 ± 6.0 vs. 163.7 ± 4.8 mmHg; $P < 0.001$). The agonistic effect on DBP was not evident in SHRs even at 12 weeks of age (sham control vs.

**A**

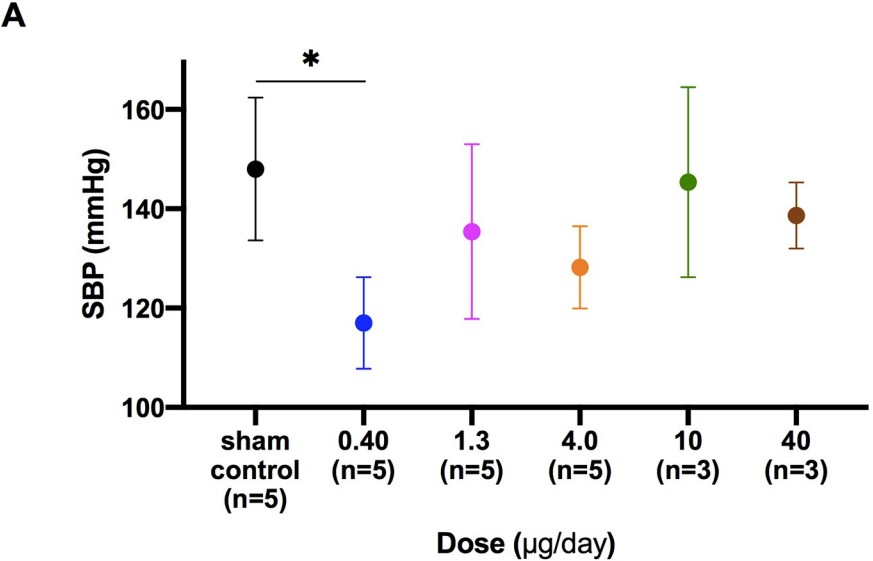

**B**

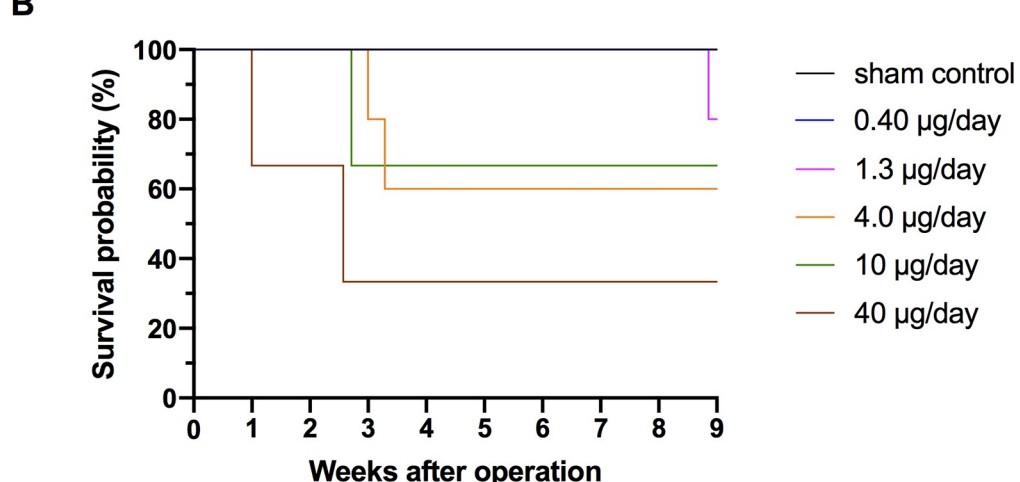

**C**

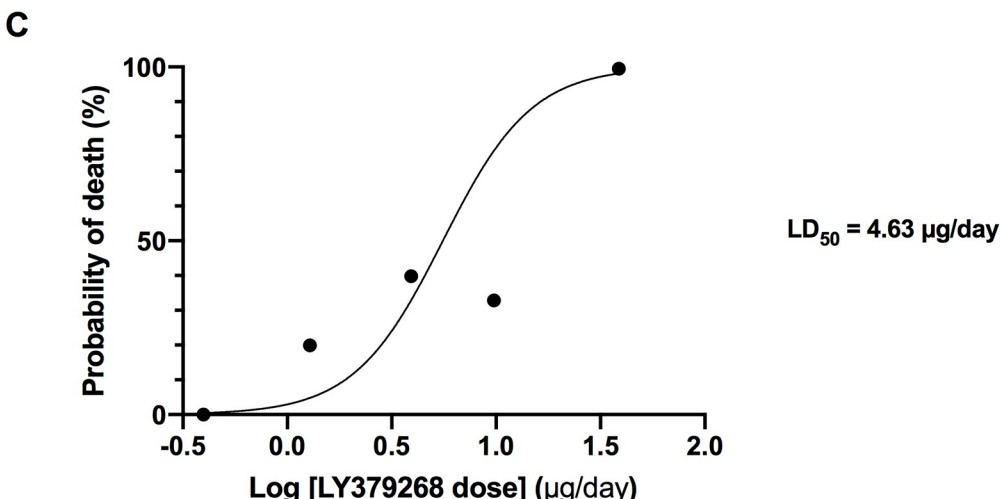

**Fig 1. SBP at 1-week post-implantation surgery and Kaplan-Meier survival estimate in response to various LY379268 doses in SHRs.** The dose of 0.40 μg/day shows the best survival ($P < 0.05$) (B) and a lower SBP compared to doses of 1.3–40 μg/day (A). Death rate at 9-week post-operation was plotted against each dose on a logarithmic scale (C). The fitted curve as a standard sigmoidal function indicates that $LD_{50}$ is approximately 4.63 μg/day. One rat that was alive at 9 weeks after administration met endpoints and was euthanized, so it was treated as 100% as a plot. $^*P < 0.05$; statistical evaluations were performed using one-way ANOVA followed by Tukey's test. Data in each group are the mean ± SEM from three or five rats. SBP, systolic blood pressure; SHR, spontaneously hypertensive rat.

agonist, 124.9 ± 21.8 vs. 116.5 ± 14.3 mmHg; $P = 1.0$) (Fig 2D). DBP decreased in SHRs at 18 weeks of age, 6 weeks after the treatment ended, but the difference was not statistically significant (sham control vs. agonist, 138.9 ± 13.1 vs. 108.1 ± 10.8 mmHg; $P = 0.082$) (Fig 2D). As

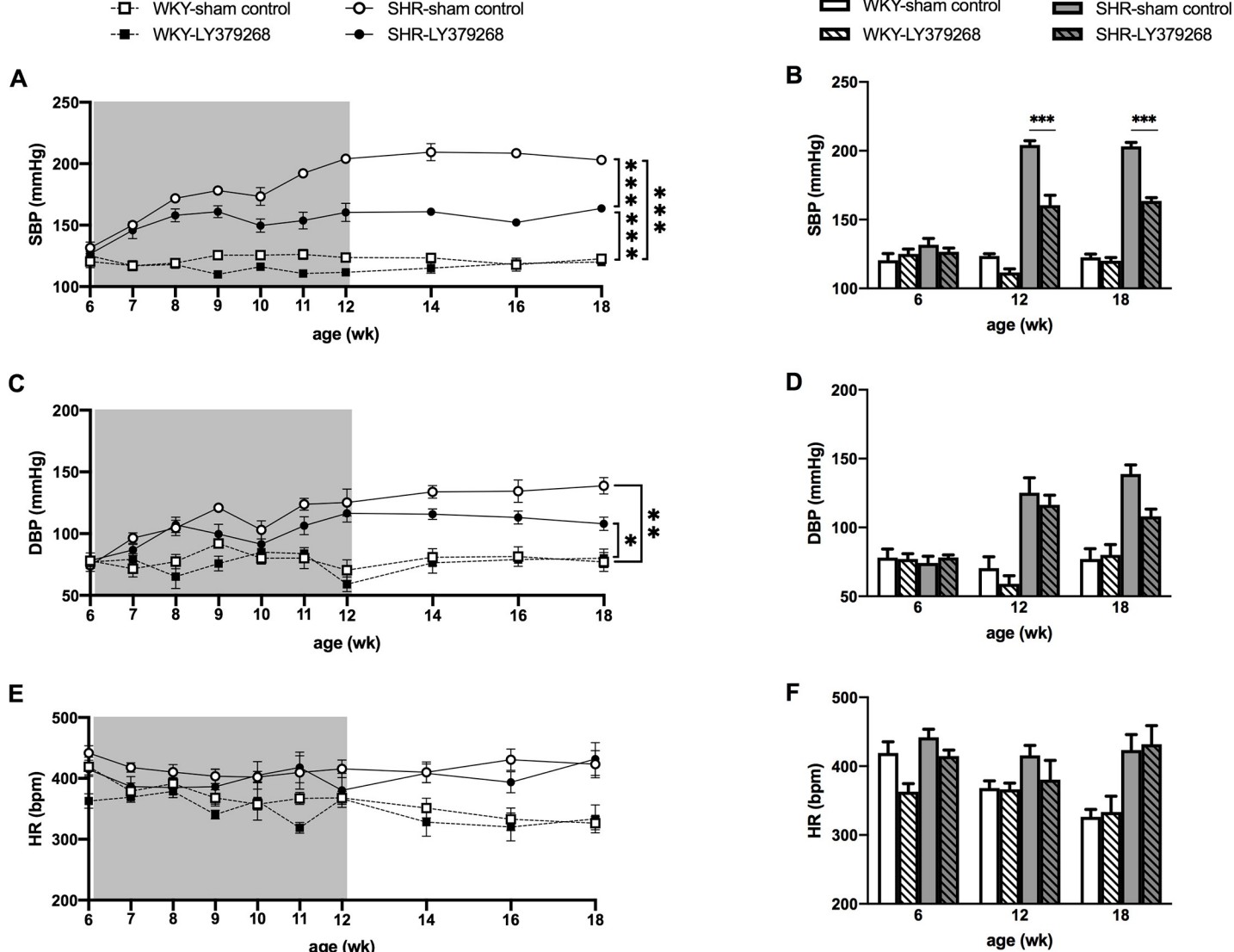

**Fig 2. Time-course changes in blood pressure and heart rate of SHRs and WKYs in the presence or absence of LY379268 (mGluR2/3 agonist) treatment.** The treatment period was 6 weeks as indicated by the shaded areas (left). The effect of LY379268 treatment on blood pressure, especially SBP, was greater in SHRs even 6 weeks post completion but was not evident in WKYs (A and B). No treatment effect was observed in DBP or HR (C-F). $^*P < 0.05$, $^{**}P < 0.01$, and $^{***}P < 0.001$; statistical evaluations were performed using two-way ANOVA followed by Tukey's test. Data for each group are the mean ± SEM from five rats. WKY, Wistar Kyoto rat; SHR, spontaneously hypertensive rat; SBP, systolic blood pressure; DBP, diastolic blood pressure; HR, heart rate; wk, week-old.

with SBP, LY379268 had no significant effect on DBP in WKYs at any age. HR was the same for all groups during the treatment period. After the treatment, HR was significantly higher in SHRs than WKYs regardless of the treatment dose (Fig 2E). The agonist effect of LY379268 on HR was not observed in either SHRs or WKYs even at 18 weeks of age (Fig 2F).

## Echocardiography and renal ultrasonography

Representative echocardiogram recordings are shown in Fig 3, and their parameters are summarized in Table 1. Neither HR nor stroke volume was changed by mGluR2/3 agonist treatment in SHRs as indicated by the similar cardiac output among groups (Table 1). In renal studies, LY379268 significantly increased the right renal artery-lowest diastolic velocity (RRA-LDV) ($P$ = 0.038) but had no effect on the left side (LRA-LDV) ($P$ = 0.62) (Table 1). Other parameters such as renal artery-peak systolic velocity, RI, and PI were not affected by LY379268 treatment in the renal artery of either side. However, the RI (sham control vs. agonist, 0.74 ± 0.04 vs. 0.68 ± 0.13) and PI (sham control vs. agonist, 1.48 ± 0.16 vs. 1.28 ± 0.46) in the right renal artery nearly reached chronic renal failure levels in the sham control group compared to the LY379268 group.

## Effects of LY379268 on HRV

HRV analysis revealed the effect of LY379268 on autonomic nervous function in conscious SHRs (Fig 4). HR decreased with aging, whereas the HRV indices of time- and frequency-domain analyses, namely, SD, CV, LF, and HF, time-dependently increased regardless of LY379268 treatment (Fig 4A). Compared with sham control, the LY379268-treated groups showed progressively increased LF (Fig 4G) and HF (Fig 4B) but decreased LF/HF ratio (Fig 4D) during the application period. The results of HRV analysis at 12 weeks of age in light and dark phases for 24 hours are presented in Fig 5. HR increased during the dark phase compared to the light phase in agonist and control groups (Fig 5A). Meanwhile, photoperiodic differences were not evident in HRV parameters obtained from either time- (SD and CV) or frequency- domain analysis (HF). LY379268 treatment had no effect on the time-domain data at 12 weeks of age (Fig 5C and 5E). For the parameters from frequency-domain analyses, especially in the light phase, HF (Fig 5B) and HFnu (Fig 5H) significantly increased, and the LF/HF ratio was significantly decreased (Fig 5D). The treatment effect on LF was completely opposite when the data were normalized by total power. A significant increase was found in LF (Fig 5G), whereas a significant decrease was observed in LFnu, especially in the light phase (Fig 5F). Given that spectral power in each range can be influenced by power in other ranges, such as the very low frequency range in this study, the latter result could be legitimate [41]. For the parameters from the frequency-domain analyses, the altered level of LF/HF ratio (Fig 5D) was dependent on changes in HF (Fig 5B), particularly in the light phase. In summary, mGluR2/3 agonist administration in the medulla oblongata simultaneously activates the parasympathetic nervous system and suppresses sympathetic nerve activity [39].

## Effect of LY379268 on catecholamine concentration in hypertensive development

The LY379268 treatment did not change the catecholamine, adrenaline (sham control vs. agonist, 1.07 ± 0.19 vs. 1.15 ± 0.32 ng/mL; n = 4 or 5 each; $P$ = 0.70), and noradrenaline (sham control vs. agonist, 0.83 ± 0.25 vs. 1.16 ± 0.42 ng/mL; n = 4 or 5 each; $P$ = 0.25) levels in the blood of SHRs (Fig 6).

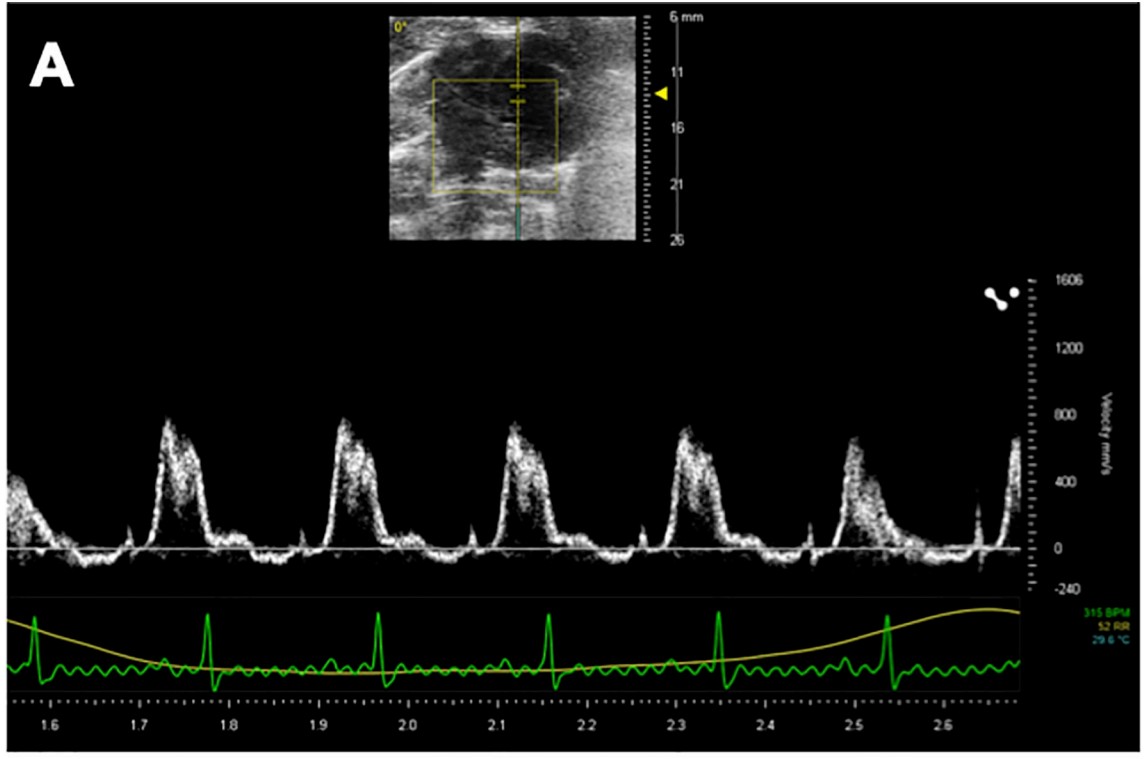

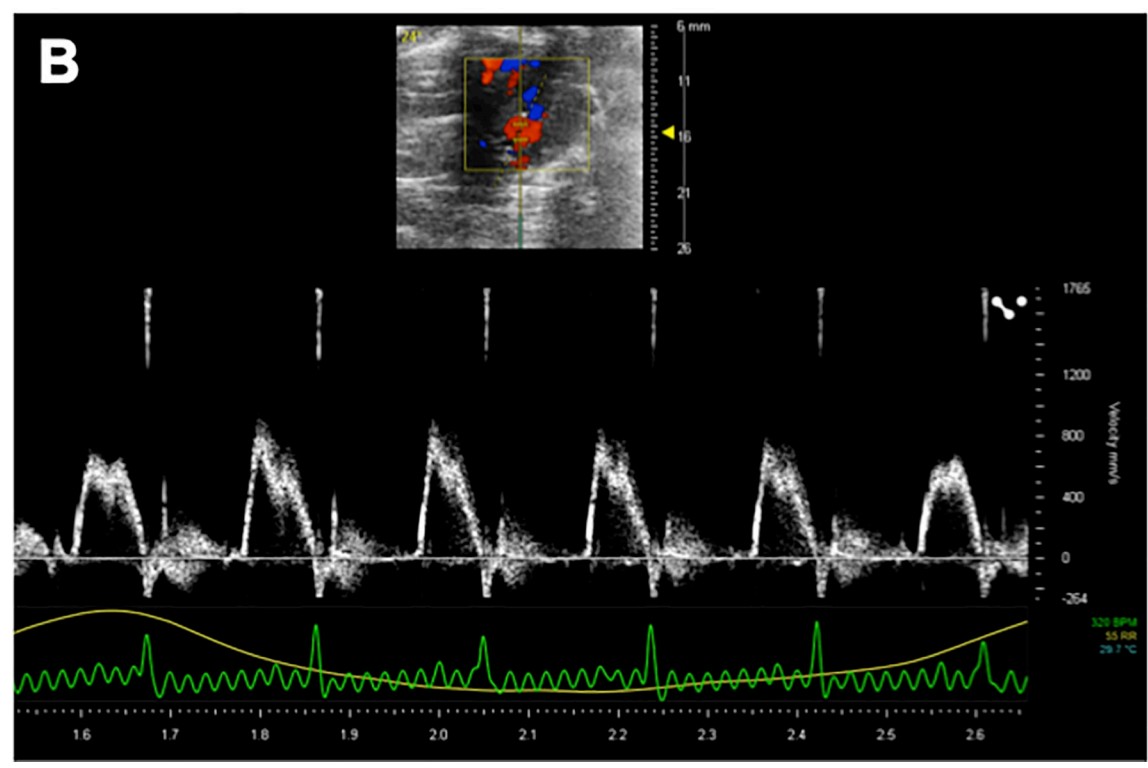

**Fig 3. Cardiac function assessed by echocardiography.** Panel A and B are representative left ventricular inflow velocity images from a sham animal and one treated with LY379268, respectively. Although the phases of the cardiac cycle are slightly different in the upper inset image, the time course changes of velocity are similar between sham and treated animals. Multiple heartbeats and respiration curves are shown in each panel's lower part.

## Effect of LY379268 on baroreflex function

The recording of HR and BP over time revealed that responses to PE or SNP exhibited a unimodal pattern (Fig 7). The initial phase had an increase in MAP (Fig 7B) and a related decrease in HR (Fig 7A) within 20 s after PE injection. The reformed phase had a sustained decrease in MAP and correlative varied HR reflex during the remaining 15-min observation period (Fig 7A and 7B). After PE injection, the MAP of both groups almost returned to the 5-min basal level, but not HR. Compared with the HR in the 5-min basal control recording, the HR of the LY379268 group was lower, while that of the sham control group was higher. After 20s from SNP injection, the primary phase showed decreased MAP (Fig 7E) and increased HR (Fig 7D). During the remaining 15-min observation period, the reformed phase had steady MAP and HR (Fig 7D and 7E). Both groups had higher level of MAP after SNP injection. The HR of the sham control group was not altered after SNP injection, while the HR of the LY379268 group showed a slight decrease. In the evaluation of baroreflex function, the LY379268 group displayed better bradycardia reflex in response to PE injection than sham control group (Fig 7C), which means that the baroreflex function of SHRs was largely augmented by LY379268 treatment. Nevertheless, both groups displayed similar levels of tachycardia reflex after SNP injection (Fig 7F).

## mGluR2/3 expression in medulla oblongata

As we stated in the introduction, we have observed the NTS neurons in SHRs showing a blunted response to mGLuR2/3 compared to those in the normotensive rats. Thus, we have

**Table 1. Effects of LY379268 in the brain stem on the cardiac and renal parameters of ultrasonography.**

| Parameter | Unit | Sham control | LY379268 |
|---|---|---|---|
| Cardiac | | | |
| Heart Rate | bpm | 284.0±16.0 | 308.1±48.3 |
| Stroke Volume | μL | 217.6±10.2 | 206.2±30.0 |
| Ejection Fraction | % | 58.1±8.4 | 65.1±5.9 |
| Fractional Shortening | % | 32.3±5.9 | 37.1±4.6 |
| Cardiac Output | mL/min | 61.7±3.5 | 64.3±15.9 |
| Renal | | | |
| RRA-PSV | mm/s | 555.9±111.4 | 813.2±447.6 |
| LRA-PSV | mm/s | 430.4±149.9 | 490.3±170.5 |
| RRA-LDV | mm/s | 143.9±22.5 | 223.6±56.0* |
| LRA-LDV | mm/s | 158.1±65.2 | 191.2±102.9 |
| RRA-RI | - | 0.74±0.04 | 0.68±0.13 |
| LRA-RI | - | 0.63±0.08 | 0.63±0.09 |
| RRA-PI | - | 1.48±0.16 | 1.28±0.46 |
| LRA-PI | - | 1.11±0.35 | 1.07±0.34 |

*$P < 0.05$ versus sham control; statistical evaluations were performed using unpaired t-test. Data in each group are mean ± SEM from four to five rats. RRA-PSV, right renal artery-peak systolic velocity; LRA-PSV, left renal artery-peak systolic velocity; RRA-LDV, right renal artery-lowest diastolic velocity; LRA-LDV, left renal artery-lowest diastolic velocity; RRA-RI, right renal artery-renal arterial resistive index; LRA-RI, left renal artery-renal arterial resistive index; RRA-PI, right renal artery-pulsatility index; LRA-PI, left renal artery-pulsatility index.

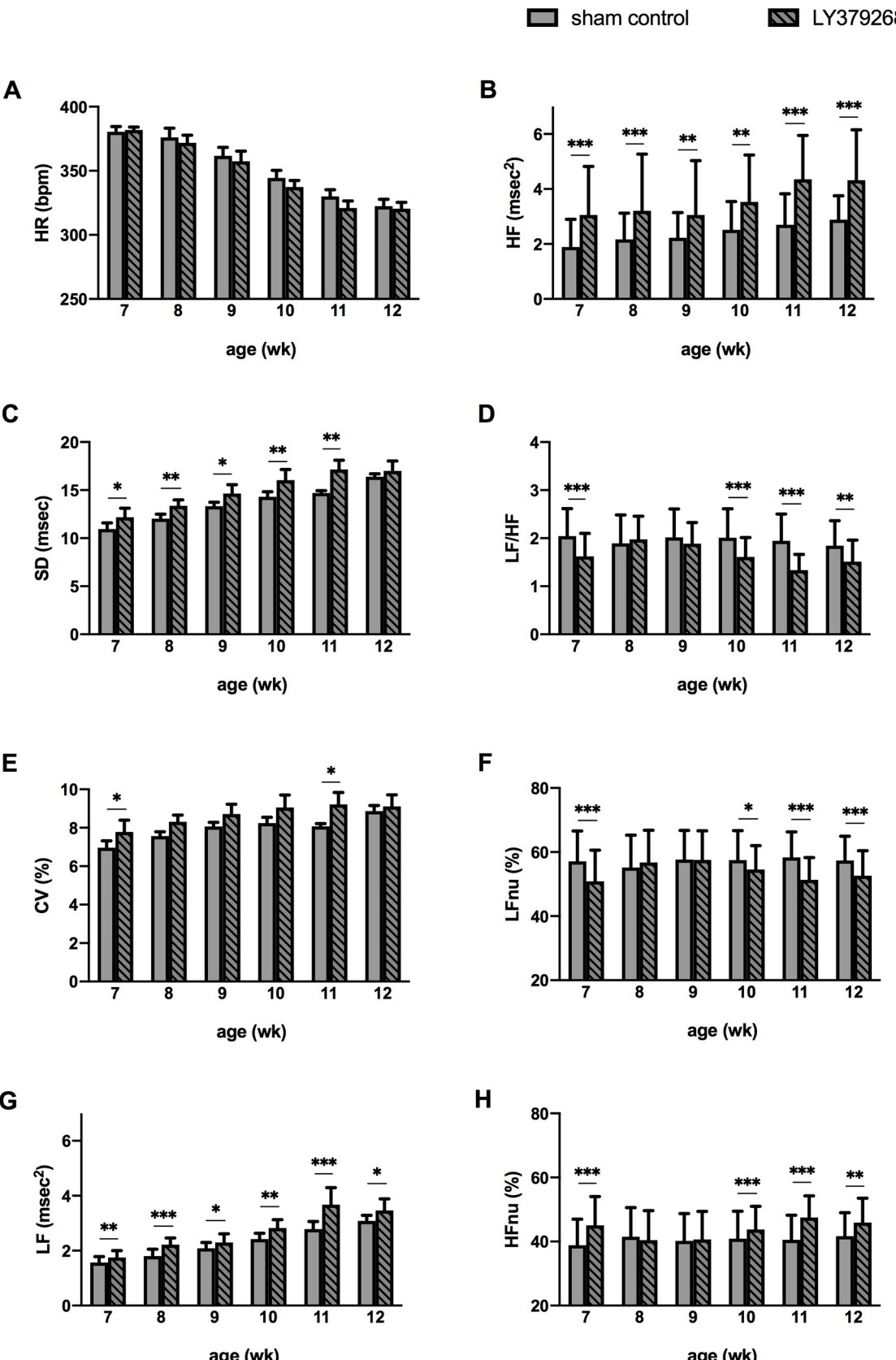

**Fig 4. Autonomic nervous function of SHR at 7–12 weeks of age during the 6-week LY379268 treatment.** LY379268 treatment increased SD (C), CV (E), LF (G), HF (B), and HFnu (H) and decreased LFnu (F) and LF/HF (D), indicating parasympathetic dominance. Normalized LF shows the opposite treatment effect. $^*P < 0.05$, $^{**}P < 0.01$, and $^{***}P < 0.001$ versus sham control; statistical evaluations were performed using unpaired t-test. Data for each group are the mean ± SEM from five rats. SHR, spontaneously hypertensive rat; HR, heart rate; SD, standard deviation; CV, coefficient of variation; LF, low frequency power; HF, high frequency power; LF/HF ratio, LF to HF ratio; LFnu, normalized low frequency power; HFnu, normalized high frequency power; wk, week-old.

checked the gene expression level of group II mGluRs between SHRs and WKYs. In sham control groups, the relative gene expression level of mGluR2 was significantly lower in SHRs (Fig 8A) compared with that in WKYs (Fig 8C). No significant difference in mGluR3 mRNA expression was found between these two strains (Fig 8B and 8D). Moreover, treatment of LY379268 did not affect mRNA expression of mGluR2 and mGluR3 in either rat strains (Fig 8).

## Discussion

This study showed that the development of hypertension was weakened by chronic mGluR2/3 agonist treatment in juvenile SHRs. Along with mGluR2/3 agonist treatment, an increase in parasympathetic nervous activity, rather than a decrease in sympathetic nervous activity, was confirmed by the relatively high HF and low LF/HF ratio from the HRV analysis. Therefore, the predominance of parasympathetic nervous activity explains the minimal increase in the BP of SHRs during and after the chronic administration of the mGluR2/3 agonist in the medulla oblongata. This finding was also supported by the lack of changes in blood catecholamine concentration regardless of the agonist treatment. Moreover, echocardiographic data showed no considerable changes in the cardiac function or renal ultrasonographic data, thus failing to support the "non-severe" hypertension effect of the agonist treatment in SHRs. The baroreflex sensitivity was improved with the mGluR2/3 agonist, particularly on reflex bradycardia. In sham control groups, the gene expression of mGluR2 in SHRs was lower than that in WKYs. All these results suggest that the enhancement of parasympathetic nervous activity via mGluR2/3-mediated stimulation may alleviate the development of hypertension and improve baroreflex function.

Although several acute experiments have already attempted to microinject mGluR modulators into the dorsal area of the medulla oblongata including the NTS [7, 13–16], no study had been conducted to date to investigate the chronic stimulation of mGluR2/3 in this brain area. Therefore, the chronic effect of mGluR2/3 stimulation on BP and the toxic effects were preliminary examined to determine the appropriate agonist dose. The results showed that the BP at 1-week post-administration was not substantially different among various concentrations. The best survival rate was observed at 0.4 μg/day, the lowest applied dose. Furthermore, this dose showed potential for chronic application because it was approximately one tenth of the 50% lethal dose ($LD_{50} = 4.63$ μg/day). Thus, 0.4 μg/day was suggested as the suitable concentration for chronical mGluR2/3 application to suppress the development of hypertension.

Hypertension development is observed in SHRs at 6–12 weeks of age [47, 48]. Given that the 6-week treatment of LY379268 blunted the development of hypertension in young SHRs, mGluR2/3 might be a possible important contributor for patients with essential hypertension. By contrast, SBP was not affected by LY379268 in normotensive rats. This finding suggests that mGluR2/3 is a silent mechanism in normal physiological conditions, especially in blood pressure regulation at the dose used in this study. mGluR involvement for blood pressure regulation has been previously reported, but the specific subtypes of mGluR responsible for this action remain unknown. The diminished SBP response induced by the mGluR agonist was

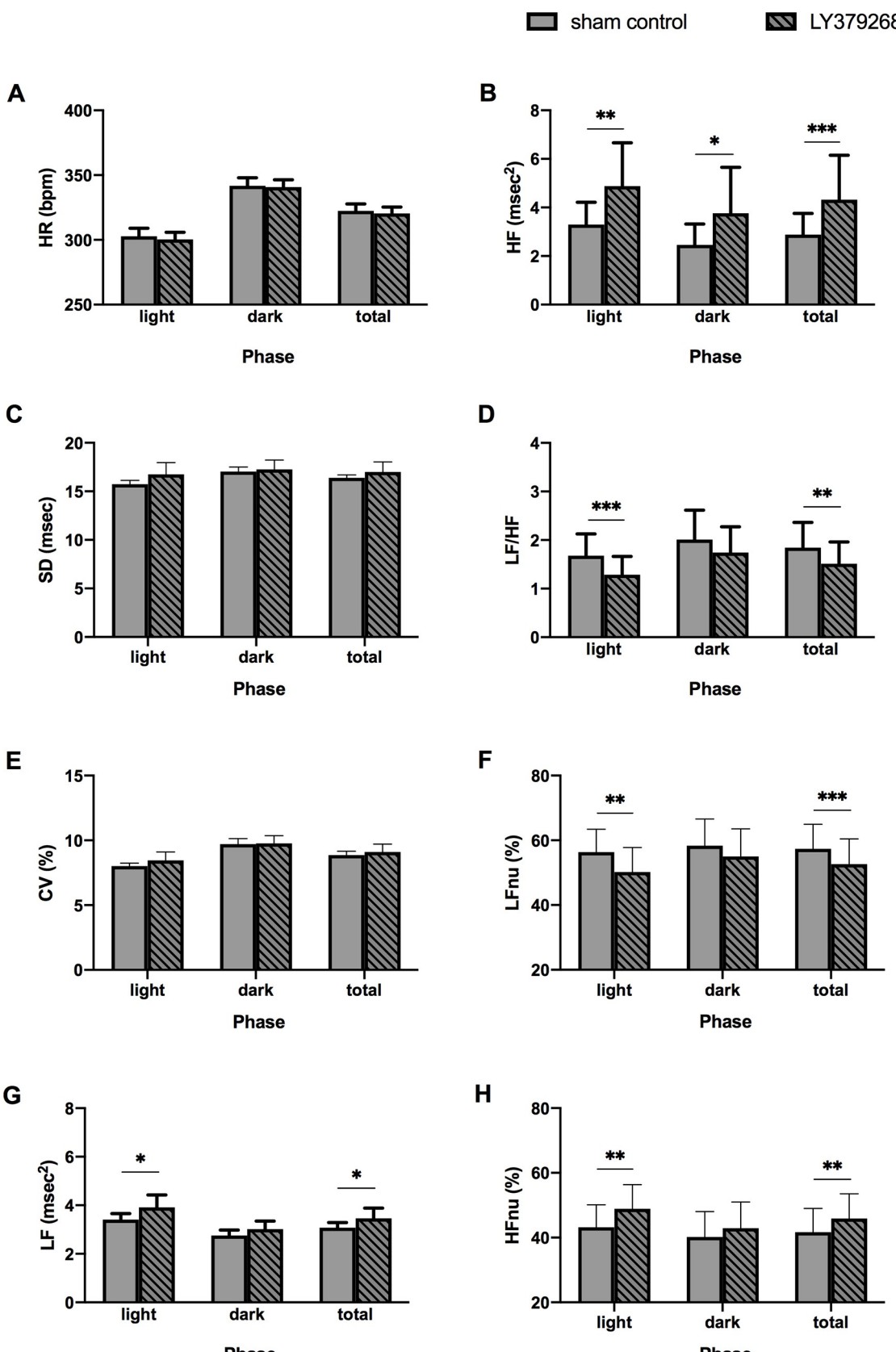

**Fig 5. Photoperiod data of autonomic nervous function in 12-week-old SHRs at the end of treatment.** Parasympathetic dominance was indicated when LY379268 treatment was completed, especially in the light phase. $^*P < 0.05$, $^{**}P < 0.01$, and $^{***}P < 0.001$ versus sham control; statistical evaluations were performed using unpaired t-test. Data for each group are the mean ± SEM from five rats. SHR, spontaneously hypertensive rat; HR, heart rate; SD, standard deviation; CV, coefficient of variation; LF, low frequency power; HF, high frequency power; LF/HF ratio, LF to HF ratio; LFnu, normalized low frequency power; HFnu, normalized high frequency power; wk, week-old.

verified with ACPD, APDC, and 4C3HPG [7, 13, 14, 16]. ACPD is a non-selective mGluR agonist that affects all subtypes of mGluR [13, 16, 49, 50]. 4C3HPG is a weak mGluR2/3 agonist but a potent antagonist of mGluR1 [50, 51]. APDC is a specific mGluR2/3 agonist with lower potency and specificity for mGluR2/3 than LY379268 [50, 51]. The advantage of this study is that the LY379268 treatment was directly administered to the dorsal medulla oblongata; hence, the response of BP and HR might originate directly from NTS, the first cardiovascular site in the dorsal medulla oblongata [5, 6]. The contribution of CVLM and RVLM which can also regulate blood pressure still cannot be excluded [6]. However, cerebrospinal fluid generally moves caudally (e.g., from the fourth ventricles to the cerebellomedullary cistern and then to the central canal) [52] and the NTS has an incomplete blood brain barrier [53], and thus drug effects may be greater at the dorsal side of medulla oblongata rather than at the ventral side where the CVLM and the RVLM located. In addition, the 6-week chronic LY379268 treatment could suppress the onset of hypertension and maintained low hypertensive levels even after its completion. To the authors' knowledge, this work is the first to study the chronic stimulation of the brainstem, including the NTS, with mGluR2/3 agonists during hypertension development. Although specific and precise targets were not determined for the agonist application, this method might be more clinically relevant than microinjection.

In our study, a slight difference of HR results between the tail-cuff method (Fig 2E) and the telemetry method (Fig 4A) was observed. The HR measurement from the tail-cuff apparatus is the pulse rate under fixation [36]. On the other hand, the HR received from the telemetry device is the results of calculation of R-R interval in unrestrained conditions [40]. The tail-cuff method also required handling for weekly measurements. Therefore, it is considered that the tail-cuff method did not show a decrease in HR throughout the experimental period because of restraint and handling.

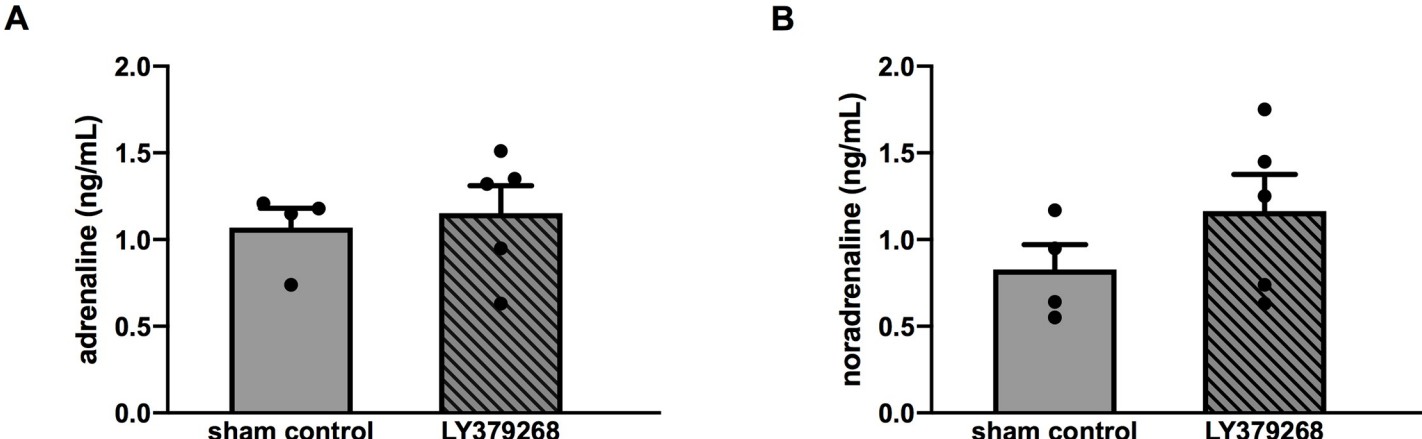

**Fig 6. Catecholamine level in 12-week-old SHRs after the completed LY379268 treatment.** No statistically significant difference in catecholamine level was found between sham control group and experimental group. Tendency of high noradrenaline concentration was detected after treatment. Statistical evaluations were performed using unpaired t-test. Data for each group are the mean ± SEM from four to five rats.

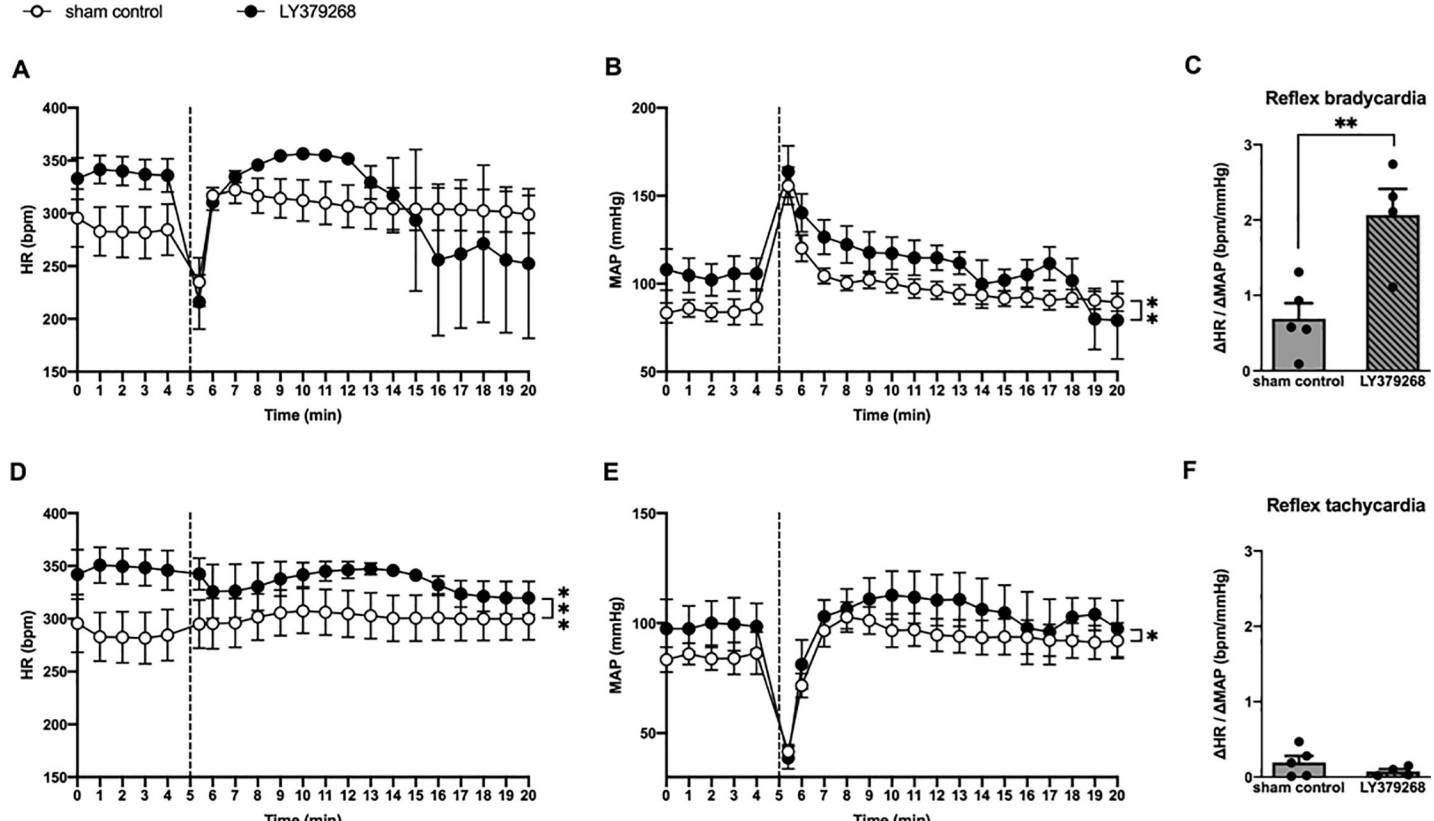

**Fig 7. Effect of LY379268 treatment on baroreflex sensitivity.** The HR and MAP response of LY379268-treated SHRs to intravenous injection of phenylephrine (21 μg/kg) (A and B) and sodium nitroprusside (50 μg/kg) (D and E) were recorded, respectively. The vertical dashed lines indicate the onset time of phenylephrine or sodium nitroprusside injection. LY379268-treated SHRs had better reflex bradycardia than sham control SHRs, but not reflex tachycardia. $^*P < 0.05$, $^{**}P < 0.01$, and $^{***}P < 0.001$ versus sham control; statistical evaluations were performed using unpaired t-test. The data in each group are mean ± SEM from four to five rats. HR, heart rate; MAP, mean arterial pressure.

The attenuation of hypertensive development is attributed to the following: First, LY379268 treatment could have altered the activity of autonomic nervous function. This notion is based on present and previous findings, in which SHRs had a higher LF level and LF/HF ratio but the same HF level as WKYs [19]. It was considered that LY379268 suppressed the SBP in WKYs depended on the lower LF/HF ratio, which represented the predominance of parasympathetic nerves activity [19, 39]. Second, because catecholamine level was unaffected by LY379268, it seemed that catecholamine was released into the blood without being affected by the parasympathetic dominance found in this study. These results suggested that mGluR2/3 activation in the medulla oblongata might hardly change peripheral resistance, including renal circulation. This finding is in good agreement with the result stating that the high BP level in the hypertensive model could be maintained by increasing vascular responsiveness or sympathetic firing [54]. SHRs have a potential impaired set point for blood pressure regulation with impaired baroreflex function, which may lead to end-organ damage [55, 56]. LY379268 may attenuate hypertensive progression by changing the balance of the sympathetic and parasympathetic nervous systems but not by reforming the set point of blood pressure regulation. Third, SD and CV are often used as indicators of baroreflex sensitivity [57]. Although the change was not throughout the chronical injection period, the increase in SD and CV in SHR

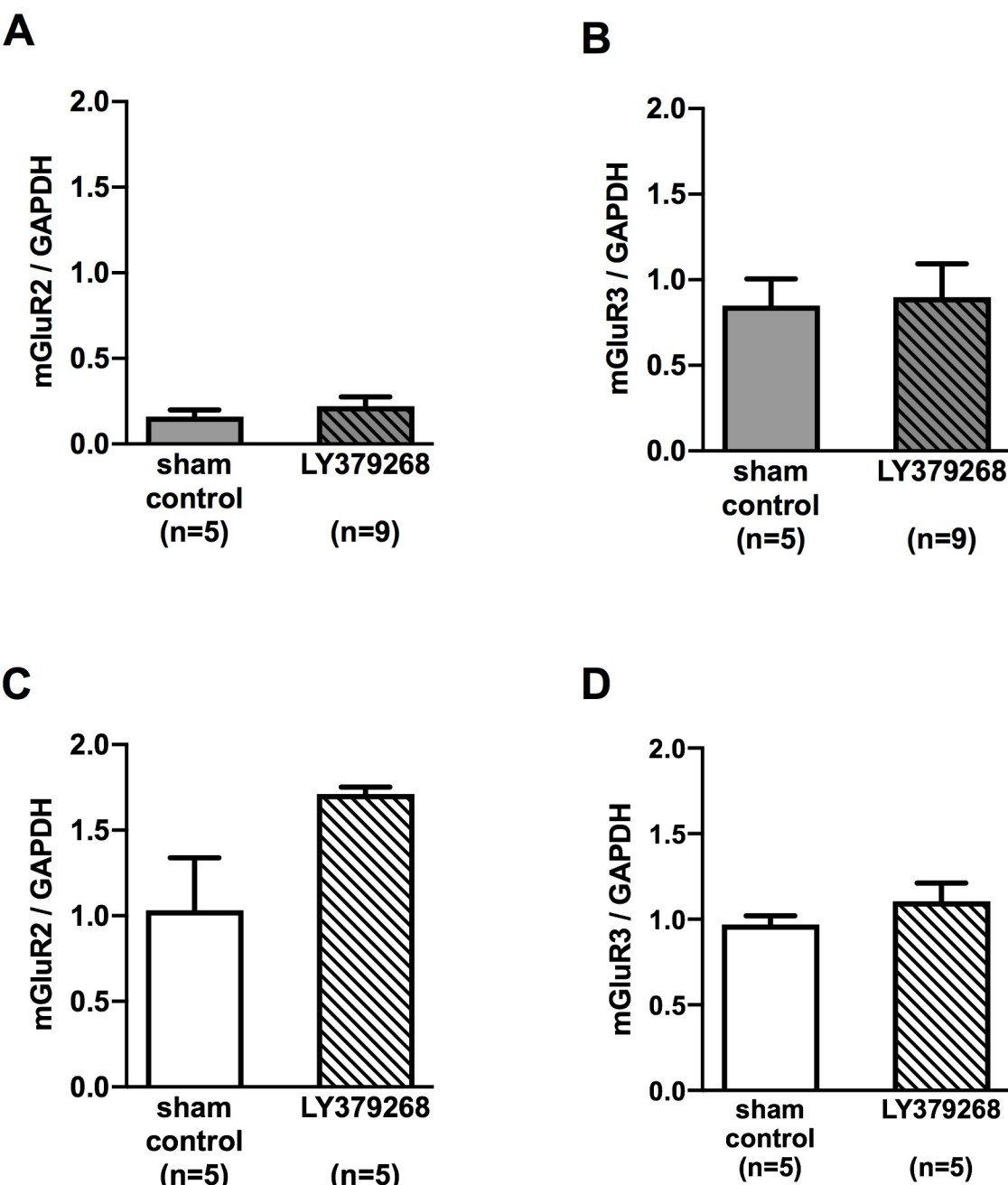

**Fig 8. Relative mRNA expression levels of mGluR2 and mGluR3 of medulla oblongata in 21-week-old sham control and LY379268 treated rats.** Each mRNA expression level was estimated by real-time PCR using GAPDH house-keeping gene expression. Greater expression of mGluR2 mRNA was observed in WKYs (C) than SHRs (A), and no difference in mGluR3 mRNA expression was observed between these two rat strains (B and D). Treatment of LY379268 did not affect mRNA expression of mGluR2 and mGluR3 in either rat strains. *$P < 0.05$; statistical evaluations were performed using unpaired t-test. Data for each group are the mean ± SEM from five rats. WKY, Wistar Kyoto rat; SHR, spontaneously hypertensive rat.

after LY379268 administration suggested the possibility of the agonist treatment improving the baroreflex function.

In terms of the baroreflex response, SHRs showed improved reflex bradycardia with LY379268 treatment, but no alteration of reflex tachycardia. LY379268 treatment on SHRs

resulted in parasympathetic dominance, which can stimulate cardiomyocytes through the vagal nerve to release acetylcholine [58]. Then, acetylcholine can bind to their $M_2$ muscarinic receptors to decrease the HR [59]. This reaction, called reflex bradycardia, was observed in the present study. In fact, Rilmenidine, a recognized antihypertensive prescription, reduces sympathetic baroreflex response and increases cardiac vagal baroreflex sensitivity [60]. No differentiation in reflex tachycardia of LY379268-treated SHRs is understandable. Reflex tachycardia usually occurs in hypotension, which is activated during low blood pressure [61]. SHRs already have the risk of developing tachycardia [62]. Although LY379268 did not change the blood's catecholamine level, it improved baroreflex function as indicated by SD and CV in HRV analysis and improved baroreflex response in reflex bradycardia.

Peripheral vascular resistance and cardiac output can easily manipulate SBP [63]. Cardiac output is mainly associated with HR and pulse pressure, which indicate the difference between SBP and DBP [63, 64]. The LY379268-treated SHRs had lower SBP than the sham control but did not show substantial alterations in their DBP and HR. Cardiac output is an index that is susceptible to changes, but the echocardiographic results in this study showed no effect of LY379268 treatment. Vascular resistance is determined by vascular structure and elastic fibers of its wall. Genetic defects on vascular structure or elastic fibers could induce congenital hypertension, and the critical timing is at fetal and early postnatal stages [63, 65]. Therefore, this mechanism may not be involved in the observed mGluR2/3 treatment effects.

SBP altercation remained in SHRs and WKYs after the treatment, suggesting that blood pressure is maintained by mGluR2/3 *per se*. The mGluR2/3 at synapses are involved in a series of plasticity processes, including synaptic depression in the NTS [12]. Thus, the synaptic plasticity of mGluR2/3 may contribute to the sustainment of altered blood pressure. On the contrary, mGluR2/3 can also be found on postsynaptic sites or cell bodies in the NTS, and their activation causes neuron membrane hyperpolarization [35]. LY379268 could inhibit the excitability of NTS neurons and thus contribute to hypertension via baroreflex pathways. In this work, the agonist prevented hypertension development to some extent. The neurons responsible for the mechanism might be the GABAergic neurons that project to higher order neurons in the autonomic nervous system.

Greater expression of mGluR2 mRNA was observed in WKYs than SHRs, although no difference in mGluR3 mRNA expression was observed between these two rat strains. Because LY379268 blunted the development of hypertension in SHRs was observed by long-term treatment, more stimulation of fewer receptors may be the effect of maintaining normal blood pressure. It has been reported that mGluR3 desensitized during 7-day administration of LY379268, while mGluR2 did not [66]. If mGluR3 desensitization persisted more than a week in the present study, observed effects were likely to be mGluR2 dependent. However, no effect of long-term LY378268 treatment was observed on the expression of both mGluR2 and mGluR3. Functional and immunohistological studies may be needed to clarify these points. Furthermore, it may also be important to investigate the quantitative and qualitative effects of endogenous agonists.

In severe progression of hypertension on SHRs, hyperactivity of the sympathetic nervous system was observed in the renal region since 2 weeks of age and consequently resulted in significant end-organ damage, including fibrohyalinosis in arterial walls and myocardial hypertrophy with thickened fibers, at the adult age [56, 67]. Given that the cardiac and renal ultrasound examination was performed around 18 weeks of age, end-organ damage was not visible in this study. Although the mechanism underlying the abnormal modulation of the cellular function of hypertensive development remains unclear, mGluR2/3 might have a specific role in blood pressure regulation. Further studies will be needed to clarify the mechanism.

In conclusion, the 6-week chronic stimulation of mGluR2/3 in the dorsal brainstem attenuates the development of hypertension by changing the activity of the autonomic nervous system involving parasympathetic dominance, which also brings better baroreflex function. The crucial mechanism involved in the attenuation of hypertensive development might be the regulation of mGluR2/3 possibly in NTS neurons. These findings may provide novel information for understanding the roles of mGluR2/3 in hypertensive development and potential therapeutic strategies. However, further research is expected to elucidate the underlying cellular mechanisms.

## Supporting information

**S1 File.**
(DOCX)

## Acknowledgments

The authors would like to thank Yoshiharu Tsuru, Research Support Department, Primetech Corp., for his technical support with electrocardiography and renal ultrasonography analyses.

## Author Contributions

**Conceptualization:** Julia Chu-Ning Hsu, Shin-ichi Sekizawa, Masayoshi Kuwahara.

**Investigation:** Julia Chu-Ning Hsu.

**Methodology:** Julia Chu-Ning Hsu.

**Supervision:** Masayoshi Kuwahara.

**Visualization:** Julia Chu-Ning Hsu.

**Writing – original draft:** Julia Chu-Ning Hsu.

**Writing – review & editing:** Shin-ichi Sekizawa, Ryota Tochinai, Masayoshi Kuwahara.

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
