## [Decision Letter · Decision Letter 0]

15 Feb 2021

PONE-D-20-38833

Therapeutic Relevance of Chronic Stimulation for Group II Metabotropic Glutamate Receptors in the Medulla Oblongata of Spontaneously Hypertensive Rats

PLOS ONE

Dear Dr. KUWAHARA,

Thank you for submitting your manuscript to PLOS ONE. After careful consideration, we feel that it has merit but does not fully meet PLOS ONE’s publication criteria as it currently stands. Therefore, we invite you to submit a revised version of the manuscript that addresses the points raised during the review process.

We look forward to receiving your revised manuscript.

Kind regards,

Luis Eduardo M Quintas, Ph.D.

Academic Editor

PLOS ONE

Journal Requirements:

2. Please amend your Ethics Statement to include information about animal anesthesia and method of sacrifice. Additionally, please amend your Methods statement to clarify if decapitation under isofluorane was the method of sacrifice used for the sham control groups.

Reviewers' comments:

Reviewer's Responses to Questions

**Comments to the Author**

1. Is the manuscript technically sound, and do the data support the conclusions?

Reviewer #1: Partly

Reviewer #2: Partly

2. Has the statistical analysis been performed appropriately and rigorously? 

Reviewer #1: I Don't Know

Reviewer #2: Yes

3. Have the authors made all data underlying the findings in their manuscript fully available?

Reviewer #1: No

Reviewer #2: Yes

4. Is the manuscript presented in an intelligible fashion and written in standard English?

Reviewer #1: Yes

Reviewer #2: Yes

5. Review Comments to the Author

Reviewer #1: The study by Hsu & cols. provides valuable and novel information on the effects of intrathecal long-term treatment with a standard mGlu2/3 agonist in hypertensive rats. The study also reveals time- and dose-dependent toxicity of chronic LY379268, which is directly relevant to the use of this compound and possibly to other mGlu2/3 agonists, either in cardiovascular or other applications. The data should be published, but the manuscript requires revision, mostly due to some overinterpretation of the results.

Datawise, a significant drawback of the study is the lack of data on mGlu2/3 expression in the LY379268-treated group (Figure 7). If possible, the authors should obtain and include these data. mGlu3 receptors desensitize during 7-day administration of LY379268, while mGlu2 receptors do not (Iacovelli et al., Mol. Pharmacol. 75:991-1003, 2009, doi: 10.1124/mol.108.052316), showing distinct regulation by agonists. Long-term treatment with the dual agonist might have significant and distinct effects in receptor expression. A paradoxical reduction of receptor expression (and activity) would not be surprising and it could challenge the authors' inference that mGlu2/3 are hypoactive and etiologically relevant in SHRs. Whatever the result, it could not only inform on the mechanism of the beneficial effect of LY379268 but also should help understanding the role of mGlu2/3 in SHR's pathophysiology.

The proposal of a selective involvement of mGlu2 (lines 409-423, 426-428 of Discussion) in SHR pathophysiology is highly speculative and seems to rely solely on its lower expression levels detected in qPCR. Since mGlu2 and mGlu3 differ in their regulation by agonist, the observed differences in mRNA could be just an epiphenomenon associated with altered glutamatergic activity in the NTS. Furthermore, there is no direct evidence that reduced expression leads to reduced activity of mGlu2, or that activity o mGlu3 is normal in SHRs. Due to the current lack of selective modulators, dissection of the receptors' functional roles would require genetic manipulations. This should be discussed and the current conclusions should be toned down accordingly.

Specific comments:

Line 3 (Title): The wording "Therapeutic relevance of..." is unnecessary.

Line 41 (Abstract): Suggested change "... by the 40 mmHg reduction in systolic blood pressure and promoted their parasympathetic ..."

Line 46 (Abstract): The term "tranquilized" seems inappropriate. In fact, it is unclear whether long-term exposure to the agonist changes up or down each receptor's activity in SHRs.

Line 61: The sentence sounds odd. Suggested change: "... (NTS) is the first synaptic site to receive afferent information from glutamatergic peripheral ..."

Lines 84-85: Previous studies on changes in mGlu2/3 expression and activity in SHR are critical to the introductory rationale and should be cited here (they were mentioned without references).

Lines 109-110: Please clarify: were the minipumps removed at 12 weeks? If not, the authors should discuss the possible impact of continued delivery during the 12-18 week period.

Line 115: Please inform the location of the implanted minipumps.

Line 186-196 (Statistical Analysis): Please clarify how the probability of death was calculated for Fig. 1C - it doesn't match the survival data of Fig. 1B. The statistical model used for the HRV time series data in Fig. 4 should be clearly indicated. Since only pairwise comparisons are shown, it is unclear whether independent t-tests or a post-hoc test (accounting for multiple testing) was used. If some type of ANOVA was used, the main effects must be reported, at least the p-value for the treatment effect (LY379268 vs. sham). These can be reported in the figure legend.

Lines 200-1: Please revise the sentence: SBP was reduced at 0.4 ug/day but tended to recover toward control values at higher doses. It was not above control at any dose.

Line 204: Please provide further details on LY379268's toxicity: what was the cause of death? Did any of the animals in the 0.4 ug/day long-term (18 w) group die? Was there any sign of distress or behavioral change?

Line 209 (Fig. 1 Legend): Please correct: contrary to what is stated, at 0.4 ug/day there was a significant hypotensive effect.

Line 222: Suggestion: "persisted" instead of "prevailed".

Line 253 (Fig. 3 Legend): Please check whether the images in A and B really represent the same phase of the cardiac cycle.

Lines 285-7: The summary interpretation of the HRV data in terms of ANS activity should be more elaborated, including references supporting it.

Lines 317-20: The qPCR experiment showing only untreated ("sham") animals requires a brief justification. It seems out of context after the description of LY379268's treatment effects.

Lines 321-6 (Fig. 7 Legend): Please clearly indicate that the experiment was with untreated (sham) animals.

Lines 329-40: This first paragraph of Discussion needs to be revised - the authors' conclusions and interpretations are presented first, then as being "confirmed" by the data. Some of the inferences seem clearly unwarranted (mGlu2 selectivity), others would require supporting references.

Lines 352-3: The statement that "LY379268 suppressed SBP gain" is not supported by the data in Fig. 2. SBP seems clearly increased in LY-treated SHRr compared to WKYs, although it was reduced compared to untreated SHRs. The authors should provide results of the appropriate statistical analysis and adjust the statement accordingly.

Lines 363-364: The authors should discuss the possibility that intrathecal LY379268 reached and affected other areas, which is expected with continuous infusion. Clearly, the effect might not be restricted to the NTS. E.g., do mGlu2/3 receptors modulate preganglionic spinal sympathetic neuron activity downstream of the RVLM?

Lines 376-80: These sentences are confusing. What is meant by "genetic aspects in peripheral areas"? The relevance to hypertension of catecholamine release from the kidney is questionable, the release of dopamine is likely to reduce BP and the cited ref. (41) is inadequate in this context.

Line 383: "hypertensive brain"?

Lines 387-9: The increases in SD and CV were at best transient - they were absent at 12 weeks (Figs. 4 and 5). This should be considered.

Lines 393-4: This sentence is confusing.

Lines 409-30: Please see general comment above and revise, avoiding undue speculation.

Reviewer #2: The manuscript reports that chronic infusion of group II mGluR2/3 (metabotropic glutamate receptor subtype 2/3) agonist into the cranial cavity near the caudal dorsal area of the medulla oblongata suppressed the development of hypertension in the SHRs (spontaneously hypertensive rat). mGluR2/3 agonist did not affect the BP in normotensive control rats (WKYs). The authors examined the changes in several parameters such as BP (by tail-cuff), heart rate variability (HRV, by telemetry ECG), cardiac and renal function (by echocardiography and renal ultrasonography), blood catecholamine concentration, and the mRNA levels of mGluR2/3 in the whole medulla oblongata. The results showed that changes in HRV might be an important mechanism. The report provides some interesting findings regarding the regulation by group II mGluR of hypertensive development in SHRs. However, I have several major concerns that should be addressed by the authors as follows.

1. Based on the introduction and abstract, NTS neurons and baroreflex regulation is the major target of the study. The application of mGluR2/3 agonist into the cranial cavity near the medulla's caudal dorsal area might diffuse out and affect many neurons. Also, the results did not show any direct evidence of the involvement of the NTS neurons (baroreflex function) in the effect of mGluR2/3 agonist on the regulation of elevated BP in SHRs. Authors should rewrite the introduction and the rationale of the study to make it more relevant to their findings. The abstract has to be changed accordingly.

2. The suppression by mGluR2/3 agonist of the hypertensive development occurs six weeks following the administration (SHR at 12 weeks of age). The authors have to explain why mGluR2 and mGluR3 mRNA levels were examined at 21-week-old rats instead of younger rats (figure 7). It is also better to determine the changes in the mRNA levels in the nuclei related to baroreflex regulation, not in the whole medulla.

3. Statistical analysis is required to analyze BP changes between SHRs and WKYs following chronic administration of the agonist at various weeks in figure 2A.

4. In figure 4, HR decreased with aging in SHRs. The change in HR is inconsistent with that in figure 2, showing no HR changes with aging in SHRs.

5. In the discussion, the authors should explain the possible mechanisms that the effects of mGluR2/3 agonists occur at chronic, but not acute, treatment.

6. Some sentences have grammatical mistakes, sometimes making it difficult to understand the meaning of statements. English editing is required. Further, it is better to cite more recent literature in the manuscript.

6. PLOS authors have the option to publish the peer review history of their article (what does this mean?). If published, this will include your full peer review and any attached files.

Reviewer #1: No

Reviewer #2: No

---

## [Author Response · Author response to Decision Letter 0]

30 Mar 2021

Dear Editor, 

Thank you very much for giving us the opportunity to revise our manuscript. We have added Ethics Statement (animal anesthesia and method of sacrifice) as suggested. We included the description about the method of sacrifice used for the sham control groups. We also appreciate both reviewers for their valuable comments. We have now carefully revised our manuscript according to the comments as indicated below.

First of all, we carried out some experiments of baroreflex sensitivity and gene expression of mGluR2/3 after LY379268 treatment, which are described in Material and Methods and Results sections in the revised manuscript. These data were newly added as Fig. 7 and Fig. 8, respectively.

Response to Reviewer #1:

The authors thank the referee for his/her valuable comments which helped to improve the manuscript. The followings are answers to the comments raised by the referee.

Reviewer’s Comments 1): 

Datawise, a significant drawback of the study is the lack of data on mGlu2/3 expression in the LY379268-treated group (Figure 7). If possible, the authors should obtain and include these data. mGlu3 receptors desensitize during 7-day administration of LY379268, while mGlu2 receptors do not (Iacovelli et al., Mol. Pharmacol. 75:991-1003, 2009, doi: 10.1124/mol.108.052316), showing distinct regulation by agonists. Long-term treatment with the dual agonist might have significant and distinct effects in receptor expression. A paradoxical reduction of receptor expression (and activity) would not be surprising and it could challenge the authors' inference that mGlu2/3 are hypoactive and etiologically relevant in SHRs. Whatever the result, it could not only inform on the mechanism of the beneficial effect of LY379268 but also should help understanding the role of mGlu2/3 in SHR's pathophysiology. 

Authors’ Response: 

As mentioned above, we added new data and cited the paper you taught us for consideration as a reference No. 66. Thank you. (Lines 377-381 of Results section and lines 499-509 of Discussion section in the revised manuscript)

Reviewer’s Comments 2): 

The proposal of a selective involvement of mGlu2 (lines 409-423, 426-428 of Discussion) in SHR pathophysiology is highly speculative and seems to rely solely on its lower expression levels detected in qPCR. Since mGlu2 and mGlu3 differ in their regulation by agonist, the observed differences in mRNA could be just an epiphenomenon associated with altered glutamatergic activity in the NTS. Furthermore, there is no direct evidence that reduced expression leads to reduced activity of mGlu2, or that activity of mGlu3 is normal in SHRs. Due to the current lack of selective modulators, dissection of the receptors' functional roles would require genetic manipulations. This should be discussed and the current conclusions should be toned down accordingly.

Authors’ Response: 

Thank you very much for the suggestion. As pointed out, we had been speculating and over-considering, so we revised the discussion into appropriate expressions throughout. (Lines 510-518 in the revised manuscript)

Specific comments:

Reviewer’s Comments 3):

Line 3 (Title): The wording "Therapeutic relevance of..." is unnecessary.

Authors’ Response: 

We have changed the title as “Chronic Stimulation of Group II Metabotropic Glutamate Receptors in the Medulla Oblongata Attenuates Hypertension Development in Spontaneously Hypertensive Rats.”

Reviewer’s Comments 4):

Line 41 (Abstract): Suggested change "... by the 40 mmHg reduction in systolic blood pressure and promoted their parasympathetic ..."

Authors’ Response: 

As suggested, we changed the phrase. Thank you. (Lines 36-40 in the revised manuscript)

Reviewer’s Comments 5):

Line 46 (Abstract): The term "tranquilized" seems inappropriate. In fact, it is unclear whether long-term exposure to the agonist changes up or down each receptor's activity in SHRs.

Authors’ Response:

Thank you for pointing this out. We have changed the sentence in a more general way. (Lines 45-47 in the revised manuscript)

Reviewer’s Comments 6): 

Line 61: The sentence sounds odd. Suggested change: "... (NTS) is the first synaptic site to receive afferent information from glutamatergic peripheral ..."

Authors’ Response:

As suggested, we changed the phrase. Thank you. (Lines 58-60 in the revised manuscript)

Reviewer’s Comments 7):

Lines 84-85: Previous studies on changes in mGlu2/3 expression and activity in SHR are critical to the introductory rationale and should be cited here (they were mentioned without references).

Authors’ Response:

Thank you for the suggestion. We have cited the appropriate publication. Since a part of knowledge was just observed during other experiments, we described so. (Lines 84-85 in the revised manuscript)

Reviewer’s Comments 8):

Lines 109-110: Please clarify: were the minipumps removed at 12 weeks? If not, the authors should discuss the possible impact of continued delivery during the 12-18 week period.

Authors’ Response:

We are sorry for not being clear. The pump was removed right after the 6-week treatment. We have described as “the pump was removed at the end of the treatment”. (Lines 110-111 in the revised manuscript)

Reviewer’s Comments 9):

Line 115: Please inform the location of the implanted minipumps.

Authors’ Response:

We added it. Thank you. (Lines 112-114 in the revised manuscript)

Reviewer’s Comments 10):

Line 186-196 (Statistical Analysis): Please clarify how the probability of death was calculated for Fig. 1C - it doesn't match the survival data of Fig. 1B. The statistical model used for the HRV time series data in Fig. 4 should be clearly indicated. Since only pairwise comparisons are shown, it is unclear whether independent t-tests or a post-hoc test (accounting for multiple testing) was used. If some type of ANOVA was used, the main effects must be reported, at least the p-value for the treatment effect (LY379268 vs. sham). These can be reported in the figure legend.

Authors’ Response:

We are sorry for not being clear. We added stats method and p-values in the text or the legend appropriately in the revised manuscript. The method of making our graph was wrong, so we remade Fig. 1C correctly. The LD50 value was no change in the reanalysis. Thank you.

Reviewer’s Comments 11):

Lines 200-1: Please revise the sentence: SBP was reduced at 0.4 ug/day but tended to recover toward control values at higher doses. It was not above control at any dose.

Authors’ Response:

Thank you for pointing this out. We have corrected the sentence as follows “systolic BP (SBP) seemed to be reduced for the 0.40 µg/day dose, but had the tendency of getting close to control values with higher doses”. (Lines 223-224 in the revised manuscript)

Reviewer’s Comments 12):

Line 204: Please provide further details on LY379268's toxicity: what was the cause of death? Did any of the animals in the 0.4 ug/day long-term (18 w) group die? Was there any sign of distress or behavioral change?

Authors’ Response:

Thank you for the suggestion. Certainly, we are not sure about the exact cause of death. However, higher dose of LY379268 seemed to let animals less active or show a less appetite. No animal died at the dose of 0.4 ug/day. We have added the general toxicity about body weight and behavior in the text. “LY379268 exhibited systemic toxicity (e.g., losing weight dramatically and less capability of action to meet endpoints for euthanasia)” (Lines 228-229 in the revised manuscript)

Reviewer’s Comments 13):

Line 209 (Fig. 1 Legend): Please correct: contrary to what is stated, at 0.4 ug/day there was a significant hypotensive effect.

Authors’ Response:

We corrected it. Thank you. (Lines 233-238 in the revised manuscript)

Reviewer’s Comments 14):

Line 222: Suggestion: "persisted" instead of "prevailed".

Authors’ Response:

As suggested, we changed it. Thank you. (Line 249 in the revised manuscript)

Reviewer’s Comments 15):

Line 253 (Fig. 3 Legend): Please check whether the images in A and B really represent the same phase of the cardiac cycle.

Authors’ Response:

We assume the upper inset images the referee was pointing out. These two images are presented at slightly different time each. However, as indicated the time course changes of velocity, no difference was observed between mGluR2/3 treatment and non-treatment. We added the sentence about the time phase “Although the phases of the cardiac cycle are slightly different in the upper inset image, the time course changes of velocity are similar between sham and treated animals”. (Lines 283-284 in revised manuscript)

Reviewer’s Comments 16):

Lines 285-7: The summary interpretation of the HRV data in terms of ANS activity should be more elaborated, including references supporting it.

Authors’ Response:

Thank you for the suggestion. We added more explanation as follows “Given that spectral power in each range can be influenced by power in other ranges, such as the very low frequency range in this study, the latter result could be legitimate [41]. For the parameters from the frequency-domain analyses, the altered level of LF/HF ratio (Fig 5D) was dependent on changes in HF (Fig 5B), particularly in the light phase. In summary, mGluR2/3 agonist administration in the medulla oblongata simultaneously activates the parasympathetic nervous system and suppresses sympathetic nerve activity [39].” (Lines 311-316 in the revised manuscript)

Reviewer’s Comments 17):

Lines 317-20: The qPCR experiment showing only untreated ("sham") animals requires a brief justification. It seems out of context after the description of LY379268's treatment effects.

Authors’ Response:

We have observed SHR showed a blunted response to mGLuR2/3 in the NTS neurons. This qPCR data could be a supportive evidence of that. We added the description in the text. “As we stated in the introduction, we have observed the NTS neurons in SHR showed a blunted response to mGLuR2/3 compared to those in the normotensive rats. Thus, we have checked the gene expression level of group II mGluRs between SHRs and WKYs.” Moreover, we added the data of LY379268 treatment as described above. (Lines 374-376, 378-381 in the revised manuscript)

Reviewer’s Comments 18):

Lines 321-6 (Fig. 7 Legend): Please clearly indicate that the experiment was with untreated (sham) animals.

Authors’ Response:

We added both sham control and LY379268 treated. Thank you. (Line 383 in the revised manuscript)

Reviewer’s Comments 19):

Lines 329-40: This first paragraph of Discussion needs to be revised - the authors' conclusions and interpretations are presented first, then as being "confirmed" by the data. Some of the inferences seem clearly unwarranted (mGlu2 selectivity), others would require supporting references.

Authors’ Response:

Thank you for the suggestion. We revised the first paragraph of Discussion. (Lines 392-406 in the revised manuscript)

Reviewer’s Comments 20):

Lines 352-3: The statement that "LY379268 suppressed SBP gain" is not supported by the data in Fig. 2. SBP seems clearly increased in LY-treated SHRs compared to WKYs, although it was reduced compared to untreated SHRs. The authors should provide results of the appropriate statistical analysis and adjust the statement accordingly.

Authors’ Response:

Thank you for the correction. We revised it. (Lines 436-438 in the revised manuscript). We have also tried to carefully describe the results with stats in Figure 2.

Reviewer’s Comments 21):

Lines 363-364: The authors should discuss the possibility that intrathecal LY379268 reached and affected other areas, which is expected with continuous infusion. Clearly, the effect might not be restricted to the NTS. E.g., do mGlu2/3 receptors modulate preganglionic spinal sympathetic neuron activity downstream of the RVLM?

Authors’ Response:

Thank you for pointing it out. We have discussed the possibility about it. (Lines 427-436 in the revised manuscript)

Reviewer’s Comments 22):

Lines 376-80: These sentences are confusing. What is meant by "genetic aspects in peripheral areas"? The relevance to hypertension of catecholamine release from the kidney is questionable, the release of dopamine is likely to reduce BP and the cited ref. (41) is inadequate in this context.

Authors’ Response:

Thank you for pointing it out. We have changed the word correctly. (Lines 454-458 in the revised manuscript)

Reviewer’s Comments 23):

Line 383: "hypertensive brain"?

Authors’ Response:

We corrected it. Thank you. (Line 459 in the revised manuscript)

Reviewer’s Comments 24):

Lines 387-9: The increases in SD and CV were at best transient - they were absent at 12 weeks (Figs. 4 and 5). This should be considered.

Authors’ Response:

Thank you for pointing it out. We added the phrase. (Lines 465-467 in the revised manuscript)

Reviewer’s Comments 25):

Lines 393-4: This sentence is confusing.

Authors’ Response:

We corrected it as follows “Cardiac output is an index that is susceptible to changes, but the echocardiographic results in this study showed no effect of LY379268 treatment.” (Lines 482-485 in the revised manuscript)

Reviewer’s Comments 26):

Lines 409-30: Please see general comment above and revise, avoiding undue speculation.

Authors’ Response:

Thank you for all the comments. Accordingly, we have revised the whole manuscript. Thank you, again.

Response to Reviewer #2:

The authors thank the referee for his/her valuable comments which helped to improve the manuscript. The followings are answers to the comments raised by the referee.

Reviewer’s Comments 1): 

Based on the introduction and abstract, NTS neurons and baroreflex regulation is the major target of the study. The application of mGluR2/3 agonist into the cranial cavity near the medulla's caudal dorsal area might diffuse out and affect many neurons. Also, the results did not show any direct evidence of the involvement of the NTS neurons (baroreflex function) in the effect of mGluR2/3 agonist on the regulation of elevated BP in SHRs. Authors should rewrite the introduction and the rationale of the study to make it more relevant to their findings. The abstract has to be changed accordingly.

Authors’ Response:

Thank you for the comments. We have discussed about the movement of cerebrospinal fluid. We agree that we cannot say that the NTS is an only mechanism for our findings here. However, we would say that baroreflex pathways are involved based on the HRV and BRS analyses. We have carefully changed the description of our manuscript and the abstract. 

Reviewer’s Comments 2):

The suppression by mGluR2/3 agonist of the hypertensive development occurs six weeks following the administration (SHR at 12 weeks of age). The authors have to explain why mGluR2 and mGluR3 mRNA levels were examined at 21-week-old rats instead of younger rats (figure 7). It is also better to determine the changes in the mRNA levels in the nuclei related to baroreflex regulation, not in the whole medulla.

Authors’ Response:

Thank you very much for pointing it out. We had an observation that NTS neurons showed less responsiveness to Group II mGluR agonist in SHR compared to the neurons in normotensive rats. Thus, we would like to show that expression level of Group II mGluR could be different between hypertensive and normotensive rats. Twenty-one week old SHRs can develop hypertension quite substantially, and thus we used this weeks of rats. We used diffusion method of mGluR2/3 agonist, and the drug was not targeting directly to the NTS. Therefore, instead of examining each nucleus in the medulla, we used the whole medulla. Certainly, the difference of mGluR subtype expression was interesting findings but the details such as aging effect would be clarified in future study. 

Reviewer’s Comments 3):

Statistical analysis is required to analyze BP changes between SHRs and WKYs following chronic administration of the agonist at various weeks in figure 2A.

Authors’ Response:

Thank you for the comment. We have changed the figure and its legend accordingly.

Reviewer’s Comments 4):

In figure 4, HR decreased with aging in SHRs. The change in HR is inconsistent with that in figure 2, showing no HR changes with aging in SHRs.

Authors’ Response:

Thank you for the comment. We have discussed the possibility of this cause of discrepancy in Lines 442-448 in the revised manuscript. 

Reviewer’s Comments 5):

In the discussion, the authors should explain the possible mechanisms that the effects of mGluR2/3 agonists occur at chronic, but not acute, treatment.

Authors’ Response:

Thank you for the suggestion. We have discussed the possible mechanism in Lines 499-509 in the revised manuscript.

Reviewer’s Comments 6):

Some sentences have grammatical mistakes, sometimes making it difficult to understand the meaning of statements. English editing is required. Further, it is better to cite more recent literature in the manuscript.

Authors’ Response:

We apologize for the poor English writing. The manuscript was edited by the highly trained English-speaking editor. We also have updated the references. Thank you.

---

## [Decision Letter · Decision Letter 1]

28 Apr 2021

Chronic Stimulation of Group II Metabotropic Glutamate Receptors in the Medulla Oblongata Attenuates Hypertension Development in Spontaneously Hypertensive Rats

PONE-D-20-38833R1

Dear Dr. KUWAHARA,

We’re pleased to inform you that your manuscript has been judged scientifically suitable for publication and will be formally accepted for publication once it meets all outstanding technical requirements. Reviewer #1 just detected a mistake in Figure 8 that should be corrected (please, see below).

Kind regards,

Luis Eduardo M Quintas, Ph.D.

Academic Editor

PLOS ONE

Additional Editor Comments (optional):

Reviewers' comments:

Reviewer's Responses to Questions

**Comments to the Author**

1. If the authors have adequately addressed your comments raised in a previous round of review and you feel that this manuscript is now acceptable for publication, you may indicate that here to bypass the “Comments to the Author” section, enter your conflict of interest statement in the “Confidential to Editor” section, and submit your "Accept" recommendation.

Reviewer #1: All comments have been addressed

Reviewer #2: All comments have been addressed

2. Is the manuscript technically sound, and do the data support the conclusions?

Reviewer #1: Yes

Reviewer #2: Yes

3. Has the statistical analysis been performed appropriately and rigorously? 

Reviewer #1: Yes

Reviewer #2: Yes

4. Have the authors made all data underlying the findings in their manuscript fully available?

Reviewer #1: Yes

Reviewer #2: Yes

5. Is the manuscript presented in an intelligible fashion and written in standard English?

Reviewer #1: Yes

Reviewer #2: Yes

6. Review Comments to the Author

Reviewer #1: Dear authors,

I am pleased with your careful consideration of my previous suggestions and I have found the revised manuscript to be significantly improved and now acceptable for publication.

In the modified Figure 8, which now contains four panels and newly added data, I noticed a small mistake that you might still want to correct:

Figure 8: figure labels A-D don't match the contents described either in the main text (lines 378-9) or in the caption (lines 385-6). In the Figure, panels A and B seem to show data for mGluR2-3 in SHR, while C and D show mGluR2-3 in WKY, based on Figure 7 of the original submission. The authors could fix this by either swapping panels B and C or their labels in the figure.

Congratulations for the interesting study.

Reviewer #2: (No Response)

7. PLOS authors have the option to publish the peer review history of their article (what does this mean?). If published, this will include your full peer review and any attached files.

Reviewer #1: **Yes: **Newton G. Castro

Reviewer #2: No

---

## [Editor Report · Acceptance letter]

10 May 2021

PONE-D-20-38833R1 

Chronic Stimulation of Group II Metabotropic Glutamate Receptors in the Medulla Oblongata Attenuates Hypertension Development in Spontaneously Hypertensive Rats 

Dear Dr. Kuwahara:

I'm pleased to inform you that your manuscript has been deemed suitable for publication in PLOS ONE. Congratulations! Your manuscript is now with our production department. 

Kind regards, 

on behalf of

Dr. Luis Eduardo M Quintas 

Academic Editor

PLOS ONE